# Cardiac Computed Tomography Angiography Plane Prediction and Comprehensive LV Segmentation

**Davis Marc Vigneault**[1] (iD)    DVIGNE01@STANFORD.EDU
[1] *Department of Radiology, Stanford University, Stanford, CA, USA*

**Ashish Manohar** [1,2,3] (iD)    ASHMAN@STANFORD.EDU
[2] *Division of Cardiovascular Medicine, Department of Medicine, Stanford University, Stanford, CA, USA*
[3] *Cardiovascular Institute, Stanford University, Stanford, CA, USA*

**Abraham Hernandez** [2]    XBRAHAM@STANFORD.EDU
**Krista Tin Chi Wong** [2]    KRISTAW@STANFORD.EDU
**Fanwei Kong** [4] (iD)    KONGF@WUSTL.EDU
[4] *Department of Mechanical Engineering and Materials Science, Washington University in St. Louis, St. Louis, MO, USA*

**Tea Gegenava** [2]    GEGENAVAT@YAHOO.COM
**Koen Nieman**[*,1,2,3]    KNIEMAN@STANFORD.EDU
**Dominik Fleischmann**[*,1,3] (iD)    D.FLEISCHMANN@STANFORD.EDU

**Editors:** Accepted for publication at MIDL 2025

## Abstract

The use of cardiac computed tomography angiography (CCTA) has dramatically increased over the past decade, with an increasingly recognized role for functional assessment; however, reformatting these datasets into standard cardiac planes and performing quantitative analysis remains time consuming and disruptive to clinical workflows. Here, we propose a fully automated, volumetric, end-to-end trained network for simultaneous detection of standard cardiac planes and comprehensive left ventricular (LV) segmentation in the predicted short axis coordinate system. The architecture consists of a coarse segmentation module, a transformation module, and a fine segmentation module. The coarse segmentation module provides an initial segmentation of the full field of view (FOV) axial images at low resolution. The transformation module predicts the rotations corresponding to the standard cardiac planes (short axis, SAX; two chamber, 2CH; three chamber, 3CH; and four chamber, 4CH) and reformats the source volume into the predicted SAX coordinate system at high resolution. Finally, the fine segmentation module segments the narrow FOV, high resolution SAX volume. The dataset consisted of 313 CCTA studies partitioned into training, validation, and testing in an 80:10:10 split. Architectural decisions are justified using ablation experiments. On the test set, the proposed architecture achieved accurate plane predictions (mean angle errors of $9.1 \pm 6.2°$, $9.5 \pm 5.4°$, $9.0 \pm 5.9°$, and $8.8 \pm 5.9°$ for the SAX, 2CH, 3CH, and 4CH planes, respectively) and high quality segmentations (Dice scores of $0.955 \pm 0.008$, $0.928 \pm 0.016$, and $0.808 \pm 0.029$ for the bloodpool, myocardium, and trabeculations, respectively). This fully automated pipeline has the potential to replace current manual workflows, expediting the availability of standard cardiac planes and quantitative analysis for clinical interpretation.

**Keywords:** Cardiac computed tomography angiography (CCTA), segmentation, spatial transformer network (STN)

---

[*] Contributed equally

## 1. Introduction

The use of cardiac computed tomography angiography (CCTA) in the United States increased 85% over the previous decade (Reeves et al., 2021), with outpatient, inpatient, and emergency department exams all more than doubling in frequency. This trend is likely to continue or accelerate owing to the increasing availability of scanners capable of performing high quality cardiac exams, incorporation of coronary CT angiography as a Class I recommendation in the AHA/ACC clinical practice guidelines on the evaluation of chest pain (Gulati et al., 2021), and doubling of reimbursement by Medicare in the United States starting in 2025 (Maxwell, 2024). Moreover, there is an increasingly recognized role of retrospectively ECG-gated cine acquisitions for functional assessment (Peper et al., 2020), with incremental value over coronary CTA alone (Seneviratne et al., 2010). Reformatting these images into standard cardiac planes is critical for standardized comparison between exams, wall thickness measurements, and myocardial segment classification; however, this processing is time consuming and usually requires a third party software package outside the standard clinical PACS system.

The literature on medical image segmentation is extensive, with deep neural networks yielding excellent performance over the past decade. Ronneberger et al. (2015) first introduced the U-Net, a highly successful 2D encoder-decoder architecture with skip connections. Since then, a plethora of modifications to the U-Net have been proposed. Residual, recurrent, and residual-recurrent versions have been described (He et al., 2016; Milletari et al., 2016; Alom et al., 2019). Oktay et al. (2018) added attention gates, using saliency maps to preserve only relevant activations. Additional connections within (Huang et al., 2017; Jegou et al., 2017) or between (Zhou et al., 2020) the network layers have been added to enhance information flow. Multiple U-Nets have been combined into "cascaded" networks, which in their simplest form provide the predictions of one U-Net module as an input to a second U-Net module (Liu et al., 2021), while more sophisticated implementations densely connect the network layers of successive U-Net modules (Wu et al., 2023). Most recently, more sophisticated approaches using transformers (Chen et al., 2021a; Cao et al., 2021) and graph neural networks (Kong et al., 2021) have also been proposed. Many of these concepts have been applied to CCTA segmentation (Bruns et al., 2020; Li et al., 2021; Jun Guo et al., 2020; Wang et al., 2022; Kong et al., 2021).

Combined segmentation and detection of standard cardiac planes from CCTA has received much less attention. The most closely related work (Chen et al., 2021b) describes a method to predict SAX, 2CH, 3CH, and 4CH planes by branching a fully connected network from a U-Net bottleneck; however, several architectural and training decisions deserve further exploration. (a) Separate models are trained to predict each cardiac plane, multiplying training time, but without comparing to a single unified model. (b) Their network is trained using a multi-stage approach, but without comparing to end-to-end training. (c) Regarding the fully connected network branched from the bottleneck, no experiments are reported exploring the effect of hidden layers (either their presence, number, or width) on performance. (d) Promising modifications to the U-Net such as attention gates and residual blocks are not explored. (e) Having learned the transformation parameters, it is reasonable to question whether performance could be improved by segmenting the reformatted images in a second stage; however, this was not investigated.

Therefore, the purpose of this study was to develop a fully automated, volumetric, end-to-end trained network for simultaneous detection of the standard cardiac planes (SAX, 2CH, 3CH, and 4CH) and comprehensive left ventricular (LV) segmentation (bloodpool, myocardium, and trabeculations) in the predicted SAX coordinate system.

## 2. Methods

### 2.1. Dataset

The dataset consisted of 313 CCTA studies randomly partitioned into training, validation, and testing in an approximately 80:10:10 split (250 training, 30 validation, and 33 testing). Cases were obtained as part of routine clinical practice and were retrospectively collected with IRB approval. Final clinical diagnoses were normal ($N = 89$), hypertrophic cardiomyopathy ($N = 106$), LV non-compaction ($N = 46$), and dilated cardiomyopathy ($N = 72$). Acquisitions were retrospectively ECG-gated and reconstructed at mid-diastole. Studies were obtained from one of four scanners: SOMATOM Force (Siemens Healthineers; $N = 275$), SOMATOM Definition Flash (Siemens Healthineers; $N = 36$), Lightspeed VCT (General Electric Healthcare; N=1), or Sensation 64 (Siemens Healthineers; $N = 1$). Slice thicknesses were 0.75 mm for Siemens and 0.625 mm for GE scans. The median reconstructed field of view (FOV) diameter was 190.0 mm (interquartile range: 173.0–209.0 mm), with a median in-plane pixel spacing of 0.37 mm (interquartile range: 0.34–0.41 mm). Initial myocardial and bloodpool segmentations were obtained using a previously described network (Kong et al., 2021) and trabeculations were separated from the bloodpool by thresholding. These initial segmentations were manually corrected (AM, AH, and KW) using ITK-Snap version 3.8.0, (Yushkevich et al., 2019). Standard cardiac planes were defined by a cardiologist with fellowship training in cardiac imaging and 10 years of experience (TG).

### 2.2. Proposed Architecture

The proposed network architecture (Figure 1) consists of three end-to-end-trained modules: (a) a coarse segmentation module, which segments a large FOV, low resolution image, (b) a transformation module, which predicts the rotations corresponding to the standard cardiac planes, and (c) a fine segmentation module, which segments a narrow FOV, high resolution image reformatted in the SAX coordinate system. Additionally, the two segmentation modules are cascaded by resampling the coarse segmentation logits and providing these as additional channels to the input of the fine segmentation module.

#### 2.2.1. Coarse Segmentation Module

The coarse segmentation module takes as input the axial CCTA volume downsampled to 3.0 mm isotropic with a $64 \times 64 \times 64$ matrix size and produces as output an equivalently sized multi-class segmentation (bloodpool, myocardium, and trabeculations). The architecture used is a volumetric attention residual U-Net with 4 downsampling/upsampling steps. The number of features produced by each convolution block is 32 in the highest resolution stage and is doubled at each downsampling step and halved at each upsampling step. The fundamental processing block is made up of a $3 \times 3 \times 3$ convolution, group normalization

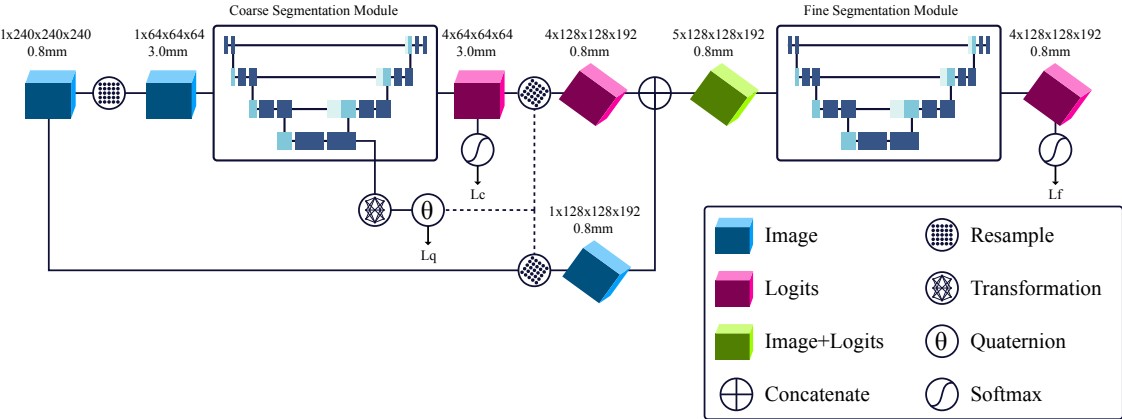

Figure 1: Proposed Network Architecture, consisting of a coarse segmentation module, a transformation module, and a fine segmentation module, trained end-to-end.

layer (Wu and He, 2018), and leaky rectified linear unit (ReLU) activation, applied twice at each stage. This is optionally converted to a residual block by element-wise addition of the input and the result of the last normalization layer, prior to applying the final activation; in practice, convolution and normalization layers are applied within the residual connection to match the feature lengths prior to summing. Traditional skip connections concatenate feature vectors from matching resolutions in the downsampling and upsampling paths. These can be converted to attention gates by first multiplying the downsampling path input by a multi-class saliency map learned from the downsampling and upsampling path inputs (Oktay et al., 2018). The result is passed into a final convolution to produce the raw logits corresponding to the background and foreground classes.

### 2.2.2. TRANSFORMATION MODULE

The transformation module is responsible for learning the rotations corresponding to the standard cardiac planes, calculating the LV centroid, and resampling the input image and the coarse segmentation logits into the learned SAX coordinate system (to be passed as input to the fine segmentation module). The SAX plane is chosen for the second segmentation stage because, unlike the long axis planes, the SAX plane is routinely reviewed as a stack from base to apex and maps directly onto the bullseye plots commonly used to display downstream analyses such as wall thickness, wall thickening, and segmental strain (Chen et al., 2023). The predicted rotations are represented as quaternions, a compact representation widely used in the graphics community due to several favorable mathematical properties. A matrix of quaternions is predicted as the output of one or more fully connected layers branched from the bottleneck of the coarse segmentation module. Note, however, that the rotations describing the standard cardiac planes ($Q_{\text{SAX}}$, $Q_{\text{2CH}}$, etc.) are the composite of (a) a shared baseline rotation ($Q_{\text{BLN}}$) orienting the long axis of the LV perpendicular to the plane of the image and (b) an additional rotation specific to each plane

($Q_{\Delta SAX}$, $Q_{\Delta 2CH}$, etc.). Therefore, rather than predicting the final rotations directly, the network is trained to predict the matrix of baseline rotation and the additional rotational offsets $[Q_{BLN}, Q_{\Delta SAX}, Q_{\Delta 2CH}, Q_{\Delta 3CH}, Q_{\Delta 4CH}]$. The LV centroid is estimated directly from the coarse segmentation prediction probabilities. Using the SAX rotation quaternion and LV centroid, the axial input image and coarse segmentation logits are resampled into the SAX coordinate system at 0.8 mm isotropic with a $128 \times 128 \times 192$ matrix size.

### 2.2.3. Fine Segmentation Module

The axial input image (and coarse segmentation logits when cascading is employed) are resampled into the SAX coordinate system at 0.8 mm isotropic with a $128 \times 128 \times 192$ matrix size and provided as input to the fine segmentation module. Like the coarse segmentation module, the fine segmentation module is a volumetric attention residual U-Net, starting with 40 features in the highest resolution stage, but otherwise identical to the former.

## 2.3. Network Implementation and Training

### 2.3.1. Preprocessing and Augmentation

The training dataset was augmented at runtime by applying random rotations (100% probability, $\pm 45°$ along each axis) and adding random Gaussian noise (50% probability, $\sigma \in [0, 100]$ HU). Note that variation in the predicted centroid and SAX quaternion consequently varies the volume provided to the fine segmentation module, resulting in additional implicit augmentation. Following augmentation, input images were clipped to the range $[-200, 600]$ Hounsfield units ("vascular windows") and normalized to the range $[0, 1]$.

### 2.3.2. Network Training

The coarse and fine segmentation modules are supervised using mean Jaccard loss across all classes ($L_c$ and $L_f$, respectively). For the transformation module, we provide both *direct* supervision of the predicted quaternions $[Q_{BLN}, Q_{\Delta SAX}, Q_{\Delta 2CH}, Q_{\Delta 3CH}, Q_{\Delta 4CH}]$ and *indirect* supervision of the composite quaternions $[Q_{SAX}, Q_{2CH}, Q_{3CH}, Q_{4CH}]$. The loss $L_q$ is the sum of the mean squared errors between the ground truth and predicted quaternions for both direct and indirect rotations, which is mathematically closely related to the angle between the rotations they represent. The total network loss $L_t$ is then given as a weighted sum of these losses:

$$L_t = \alpha_c L_c + \alpha_q L_q + \alpha_f L_f \tag{1}$$

We set $\alpha_c = \alpha_f = 1$ and $\alpha_q = 10/n_q$ where $n_q$ is the total number of quaternions being supervised. Additional training and implementation details are given in Appendix A.

## 3. Experiments and Results

Results of the hyperparameter search and ablation experiments are presented in Table 1 (angle errors) and Table 2 (centroid errors and Dice scores). Regarding the transformation module, the depth and width of the hidden layers branched from the coarse segmentation module bottleneck were varied. Among these, the version with two 128-feature hidden layers

(abbreviated "128-128") performed best in terms of centroid error ($0.805 \pm 0.521$mm), angle error for three of the four standard cardiac planes ($9.1 \pm 6.2°$ SAX, $9.0 \pm 5.9°$ 3CH, and $8.8 \pm 5.9°$ 4CH), and angle error for the baseline rotation $Q_{\mathrm{BLN}}$ ($6.6 \pm 3.7°$). For the 2CH plane, the angle error was similar between the 128-128 and best performing networks. Regarding segmentation performance, the 128-128 network performed slightly worse compared to the best performing network in terms of myocardial Dice ($0.928 \pm 0.016$ vs $0.930 \pm 0.016$, $p < 0.05$) and trabeculation Dice ($0.808 \pm 0.029$ vs $0.814 \pm 0.030$, $p < 0.05$). Bloodpool Dice was similar between the 128-128 and best performing networks. Because the 128-128 network performed best overall in predicting the standard cardiac planes and differences in Dice score compared to the best performing networks were small, the 128-128 network was selected as the baseline for subsequent ablation experiments; representative segmentations and plane predictions are shown in Figure 2.

Ablation experiments were performed to explore the value of attention gates, residual blocks, cascading, indirect and direct rotation supervision, end-to-end training, the fine segmentation module, multiple vs single plane predictions, and hidden layers in the transformation module. Metrics which demonstrated a statistically significant change compared to the proposed network by paired Student's $t$-test ($\alpha = 0.05$) are reported below. Removing the attention gates degraded performance in terms of centroid error but improved trabeculation Dice ($0.813 \pm 0.029$ versus $0.808 \pm 0.029$, $p < 0.05$). Removing residual blocks degraded performance for bloodpool Dice. Removing cascading (that is, providing only the resampled input image without the coarse segmentation logits to the fine segmentation module) degraded performance in terms of the baseline rotation $Q_{\mathrm{BLN}}$ but improved trabeculation Dice ($0.817 \pm 0.029$ versus $0.808 \pm 0.029$, $p < 0.05$). Removing indirect supervision of the quaternion rotations degraded performance in terms of the angle errors for all standard cardiac planes and for myocardial Dice. Removing direct supervision of the quaternion rotations degraded performance in terms of the baseline rotation $Q_{\mathrm{BLN}}$.

To test the effect of end-to-end training, we sequentially trained the coarse segmentation, transformation, and fine segmentation modules (8 epochs each, 24 epochs total), resulting in degraded performance for all metrics. To test the utility of our two-stage segmentation approach, we removed the fine segmentation module, instead inputting the full field of view, high-resolution images to the first segmentation stage (requiring a reduction in the number of features in the first stage U-Net by a factor of 4 due to GPU memory constraints). Doing so degraded performance in terms of the 3CH and 4CH angle errors, but improved bloodpool Dice ($0.958 \pm 0.008$ vs $0.955 \pm 0.008$, $p < 0.05$) and trabeculation Dice ($0.834 \pm 0.029$ vs $0.808 \pm 0.029$, $p < 0.05$). To test the utility of predicting all standard cardiac planes in a single network, we trained four separate networks, each predicting a single cardiac plane, following the approach taken by Chen et al. (2021b). Note that the input image and coarse segmentation logits were resampled into whichever clinical plane was predicted, as the SAX rotation was not always available. The SAX-, 2CH-, and 4CH-only networks all exhibited degraded performance in terms of bloodpool and trabeculation Dice. The 3CH-only network was not significantly different in terms of any metric. Finally, removing all hidden layers from the transformation module degraded performance in terms of angle errors for the baseline rotation $Q_{\mathrm{BLN}}$, SAX, 2CH, and 3CH planes, and additionally degraded performance in terms of bloodpool Dice.

Table 1: Cardiac plane angle errors. The "An" (attention gates), "Rs" (residual blocks), "Cd" (cascading), "Id" (indirect rotation supervision), "Dr" (direct rotation supervision), "EE" (end-to-end training), and "Fn" (fine segmentation module) columns indicate whether the feature was ("+") or was not ("−") employed. The "Pn" (plane) column indicates whether the model was trained to predict "All" planes or a single ("SAX", "2CH", "3CH", or "4CH") plane. The "Hn" (hidden layer) column indicates the number of features in each hidden layer of the transformation module (e.g., "64": one 64-feature hidden layer; "64-64": two 64-feature hidden layers; "−": no hidden layers). Values are reported as "mean ± standard deviation". Results significantly improved and worsened relative to the proposed network (highlighted in gray) are highlighted in green and red, respectively.

| Network Parameters | | | | | | | | | Angle Error (°) | | | | |
| An | Rs | Cd | Id | Dr | EE | Fn | Pn | Hn | BLN | SAX | 2CH | 3CH | 4CH |
| --- | --- | --- | --- | --- | --- | --- | --- | --- | --- | --- | --- | --- | --- |
| + | + | + | + | + | + | + | All | 64 | 7.4±3.7 | 9.7±5.8 | 9.9±5.7 | 9.8±5.2 | 9.6±5.1 |
| + | + | + | + | + | + | + | All | 64-64 | 7.3±3.7 | 10.1±5.6 | 10.0±5.3 | 10.7±5.4 | 9.4±5.4 |
| − | + | + | + | + | + | + | All | 128 | 6.8±3.4 | 9.2±5.8 | 9.1±6.5 | 9.9±5.5 | 10.9±7.5 |
| + | + | + | + | + | + | + | All | 128-128 | 6.6±3.7 | 9.1±6.2 | 9.5±5.4 | 9.0±5.9 | 8.8±5.9 |
| + | + | + | + | + | + | + | All | 256 | 7.6±3.5 | 9.7±5.5 | 10.3±5.0 | 10.7±4.9 | 10.0±5.5 |
| + | + | + | + | + | + | + | All | 256-256 | 6.9±3.0 | 10.0±5.0 | 9.7±5.2 | 9.7±4.8 | 9.3±5.2 |
| − | + | + | + | + | + | + | All | 128-128 | 6.9±3.5 | 9.9±4.4 | 9.9±4.9 | 9.7±4.9 | 9.2±4.8 |
| + | − | + | + | + | + | + | All | 128-128 | 6.9±3.1 | 9.3±5.4 | 9.6±5.5 | 9.8±5.5 | 9.2±6.0 |
| + | + | − | + | + | + | + | All | 128-128 | 7.6±3.6 | 10.1±4.9 | 9.4±6.0 | 10.0±4.8 | 9.3±4.9 |
| + | + | + | − | + | + | + | All | 128-128 | 7.4±3.6 | 10.8±5.5 | 10.6±6.2 | 10.6±5.5 | 10.8±5.4 |
| + | + | + | + | − | + | + | All | 128-128 | 182.7±4.2 | 9.4±5.6 | 9.0±5.8 | 9.1±5.8 | 8.8±5.9 |
| + | + | + | + | + | − | + | All | 128-128 | 9.7±5.0 | 13.0±5.9 | 12.5±7.0 | 12.8±5.4 | 12.5±5.7 |
| + | + | + | + | + | + | − | All | 128-128 | 7.6±3.5 | 11.1±5.9 | 10.9±6.6 | 10.4±5.6 | 10.6±6.3 |
| + | + | + | + | + | + | + | SAX | 128-128 | 6.6±3.6 | 8.7±5.0 | | | |
| + | + | + | + | + | + | + | 2CH | 128-128 | 6.2±3.6 | | 8.7±5.6 | | |
| + | + | + | + | + | + | + | 3CH | 128-128 | 6.9±3.4 | | | 9.0±4.4 | |
| + | + | + | + | + | + | + | 4CH | 128-128 | 7.1±3.6 | | | | 9.0±5.8 |
| + | + | + | + | + | + | + | All | — | 8.1±3.9 | 11.6±7.1 | 10.8±6.7 | 10.4±6.1 | 10.1±6.0 |

## 4. Discussion and Conclusions

Here, we present a fully automated, volumetric, end-to-end trained network for simultaneous detection of standard cardiac planes (SAX, 2CH, 3CH, and 4CH) and comprehensive LV segmentation (bloodpool, myocardium, and trabeculations) in the predicted SAX coordinate system. The network had high performance in terms of standard cardiac plane detection, with sub-millimeter centroid error and angle error $< 10°$ for all standard cardiac planes. The Dice scores achieved by our network are also high ($0.955 \pm 0.008$, $0.928 \pm 0.016$, and $0.808 \pm 0.029$ for the bloodpool, myocardium, and trabeculations, respectively), which is notable given the separate segmentation of the LV trabeculations, a high surface-area-to-volume structure. Note that segmentation of the LV trabeculations has value in the investigation of diagnoses such as LV non-compaction cardiomyopathy (Manohar et al., 2023) but is not typically included as a separate label in segmentation models.

Several key points may be gleaned from our ablation experiments. First, end-to-end training resulted in significantly improved performance for all metrics compared to training each module separately for a fixed total number of epochs. Second, training separate models to predict each cardiac plain individually–the approach taken in Chen et al. (2021b)–failed to significantly improve angle errors, in spite of quadrupling the total training time required compared to our single unified model. Third, providing direct supervision of the quaternions, while not significantly changing the final composite rotations or segmentation performance, was necessary to provide accurate intermediate rotations, which are useful in the event that the predicted planes require manual correction. Fourth, we found that the number and width of hidden layers in the transformation module was an important hyperparameter with significant impact on network performance.

This work has several limitations and areas for future improvement and validation. First, it would be useful to quantify intra- and inter-observer variability in standard cardiac plane angles in order to contextualize the angle errors observed in our network. Second, several potential improvements to the segmentation modules, particularly the use of transformer-based modules, have the potential to improve segmentation performance and should be investigated. Third, whereas our intention in adding cascading (passing features from the coarse segmentation module to the fine segmentation module) was to improve segmentation performance, removing cascading instead resulted in significantly *higher* angle error for the baseline rotation and slightly *higher* trabeculation Dice; this paradoxical result is not fully explained by our experiments and is deserving of further investigation. Fourth, removing the fine segmentation module degrades 3CH and 4CH cardiac plane prediction while very slightly *improving* bloodpool and trabeculation Dice; these somewhat counterintuitive results also deserve further investigation. Fifth, the dataset was obtained retrospectively from a single center; proposed network should undergo further validation in prospectively obtained, multicenter images. Last, although we explore through ablation experiments many of the features which distinguish our network from the most closely related work Chen et al. (2021b), a direct head-to-head comparison would be valuable.

This fully automated pipeline has the potential to replace current manual workflows, expediting the availability of standard cardiac planes and quantitative analysis for interpretation.

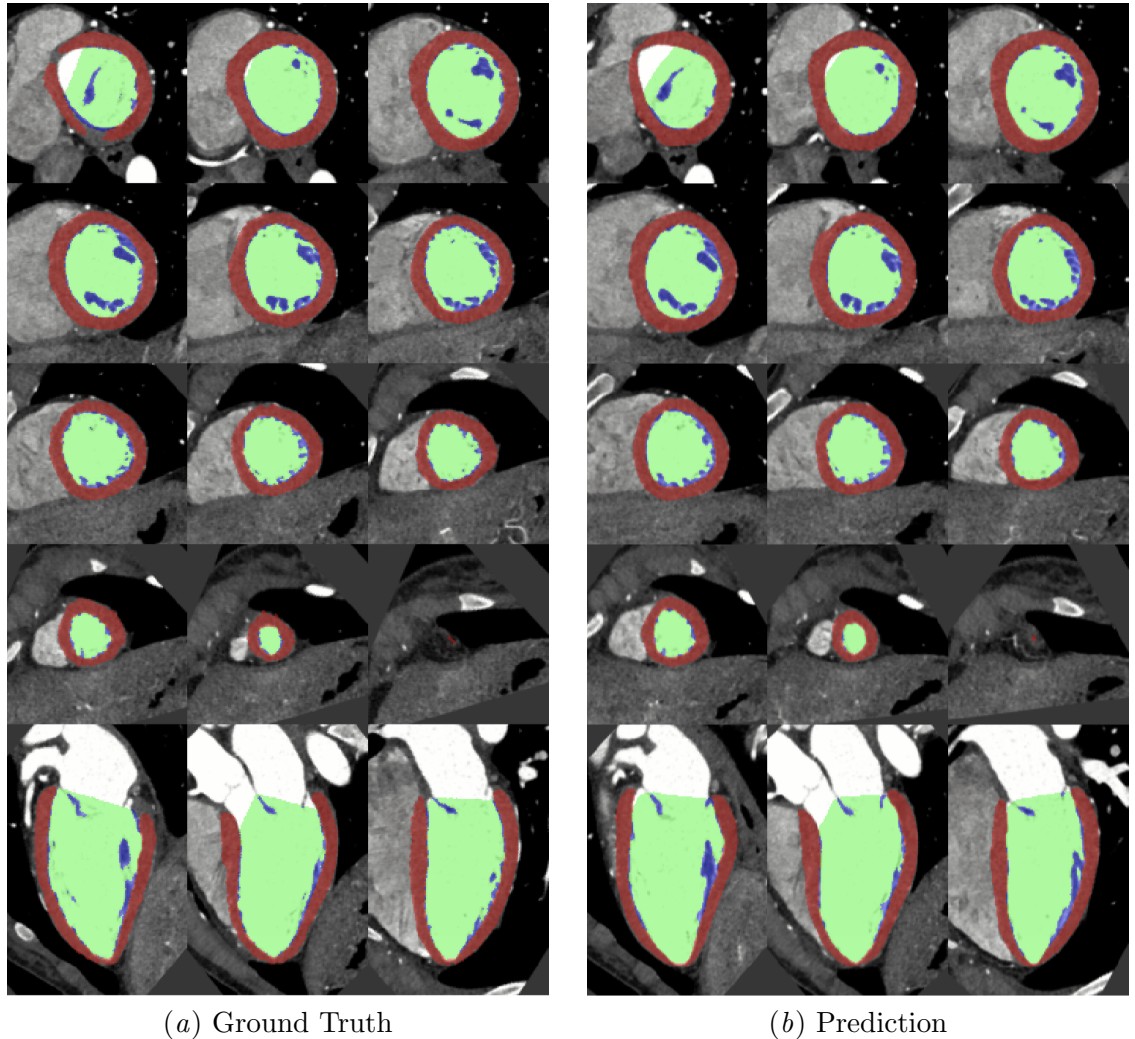

(*a*) Ground Truth          (*b*) Prediction

Figure 2: Representative ground truth (left) and predicted (right) segmentations and standard cardiac planes for a patient in the testing partition. In each case, the SAX stack is shown in rows 1-4 (sliced from base to apex) and the long axis reformations are shown in row 5 (2CH, 3CH, and 4CH from left to right).

## Acknowledgments

This study was supported by grants from the Radiological Society of North America (RR24-065; DV), the Etta K. Moskowitz Foundation (DV), the American Heart Association (AHA 24POST1187968; AM), and the National Institutes of Health (NHLBI R01 HL146754; KN).

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

## Appendix A. Implementation Details

The network was trained end-to-end with a batch size of 1 for 24 epochs using the Adam optimizer. Learning rate warmup was used with an initial rate of $10^{-6}$ and a target rate of $10^{-4}$, achieved using a linear ramp over 3 epochs. After the third epoch, the learning rate was exponentially decayed with a multiplicative factor of 0.9.

The network was implemented in python (version 3.12.6) using Monai (version 1.5) and PyTorch (version 2.5.1+cu124). Conversion between quaternion and matrix rotation representations was performed using RoMa (version 1.5.0). Experiments were run on an Ubuntu workstation (version 24.04) with a 16 core Intel i7-13700K processor, 64 GB RAM, and a single NVIDIA GeForce RTX 4090 GPU with 24 GB memory. Please see the repository for additional details.[1]

---

1. https://github.com/sudomakeinstall/2025-midl-ccta-plane-prediction

Table 2: Centroid errors and Dice scores. The "An" (attention gates), "Rs" (residual blocks), "Cd" (cascading), "Id" (indirect rotation supervision), "Dr" (direct rotation supervision), "EE" (end-to-end training), and "Fn" (fine segmentation module) columns indicate whether the feature was ("+") or was not ("–") employed. The "Pn" (plane) column indicates whether the model was trained to predict "All" planes or a single ("SAX", "2CH", "3CH", or "4CH") plane. The "Hn" (hidden layer) column indicates the number of features in each hidden layer of the transformation module (e.g., "64": one 64-feature hidden layer; "64-64": two 64-feature hidden layers; "–": no hidden layers). Values are reported as "mean ± standard deviation". Results significantly improved and worsened relative to the proposed network (highlighted in gray) are highlighted in green and red, respectively.

| Network Parameters | | | | | | | | | Error (mm) | Dice | | |
| An | Rs | Cd | Id | Dr | EE | Fn | Pn | Hn | Centroid | BP | MC | TB |
|----|----|----|----|----|----|----|-----|--------|-------------|-------------|-------------|-------------|
| + | + | + | + | + | + | + | All | 64 | 2.246±1.116 | 0.955±0.007 | 0.930±0.016 | 0.814±0.030 |
| + | + | + | + | + | + | + | All | 64-64 | 2.184±1.077 | 0.956±0.007 | 0.929±0.014 | 0.810±0.032 |
| + | + | + | + | + | + | + | All | 128 | 0.853±0.554 | 0.955±0.007 | 0.926±0.016 | 0.808±0.029 |
| + | + | + | + | + | + | + | All | 128-128 | 0.805±0.521 | 0.955±0.008 | 0.928±0.016 | 0.808±0.029 |
| + | + | + | + | + | + | + | All | 256 | 1.941±1.043 | 0.955±0.007 | 0.930±0.016 | 0.809±0.032 |
| + | + | + | + | + | + | + | All | 256-256 | 0.820±0.605 | 0.954±0.007 | 0.921±0.016 | 0.801±0.032 |
| – | + | + | + | + | + | + | All | 128-128 | 1.917±1.000 | 0.954±0.007 | 0.928±0.016 | 0.813±0.029 |
| + | – | + | + | + | + | + | All | 128-128 | 0.811±0.575 | 0.954±0.007 | 0.927±0.014 | 0.805±0.031 |
| + | + | – | + | + | + | + | All | 128-128 | 0.787±0.562 | 0.957±0.008 | 0.931±0.017 | 0.817±0.029 |
| + | + | + | – | + | + | + | All | 128-128 | 0.781±0.629 | 0.954±0.007 | 0.925±0.016 | 0.805±0.031 |
| + | + | + | + | – | + | + | All | 128-128 | 0.772±0.492 | 0.955±0.007 | 0.929±0.015 | 0.812±0.029 |
| + | + | + | + | + | – | + | All | 128-128 | 1.302±0.739 | 0.927±0.013 | 0.905±0.017 | 0.717±0.041 |
| + | + | + | + | + | + | – | All | 128-128 | 0.800±0.599 | 0.958±0.008 | 0.926±0.019 | 0.834±0.029 |
| + | + | + | + | + | + | + | SAX | 128-128 | 0.795±0.565 | 0.950±0.009 | 0.927±0.017 | 0.799±0.040 |
| + | + | + | + | + | + | + | 2CH | 128-128 | 0.864±0.562 | 0.951±0.008 | 0.927±0.016 | 0.801±0.038 |
| + | + | + | + | + | + | + | 3CH | 128-128 | 0.830±0.629 | 0.954±0.008 | 0.929±0.016 | 0.810±0.034 |
| + | + | + | + | + | + | + | 4CH | 128-128 | 0.764±0.539 | 0.950±0.010 | 0.926±0.015 | 0.797±0.042 |
| + | + | + | + | + | + | + | All | – | 0.801±0.445 | 0.954±0.007 | 0.926±0.014 | 0.805±0.033 |

