# OpenReview forum: "Cardiac Computed Tomography Angiography Plane Prediction and Comprehensive LV Segmentation"
_MIDL.io/2025/Conference — MIDL 2025 Poster_

### Official Review · Reviewer_sMe4 · 2025-02-20

**Confidence:** 2
**Preliminary Rating:** 4
**Recommendation:** Best Paper Award, Oral, Poster

**Summary:**

This paper presents a fully automated deep learning pipeline for cardiac plane prediction and left ventricular (LV) segmentation from CCTA scans. The proposed model efficiently integrates standard cardiac plane detection and high-resolution segmentation, improving clinical workflow efficiency.

**Strengths:**

The study introduces a novel volumetric deep learning architecture that simultaneously predicts cardiac planes and performs comprehensive LV segmentation, reducing manual workload. The model achieves impressive Dice scores (0.955 for bloodpool, 0.928 for myocardium, and 0.808 for trabeculations), ensuring reliable cardiac structure delineation. The use of quaternions for cardiac plane rotation estimation results in precise reorientation, with mean angular errors below 10° for all standard cardiac planes.

**Weaknesses:**

The model was trained on a single-center retrospective dataset, which may limit its applicability in diverse clinical settings. Future validation on multi-center, prospective datasets is necessary to confirm robustness.

**Detailed Comments:**

Nan

**Justification Of The Preliminary Rating:**

Although I am not an expert in the medical field, from a machine learning perspective, I believe this work is publishable. The reviewer understands the difficulty of data collection and hopes to see more comprehensive data in your future work.

**Questions To Address In The Rebuttal:**

Nan

**Special Issue:**

Yes

---

> ### Author Response · Authors · 2025-03-08
> **Response to Reviewer sMe4**
>
> - R3-1: "The model was trained on a single-center retrospective dataset, which may limit its applicability in diverse clinical settings. Future validation on multi-center, prospective datasets is necessary to confirm robustness."
>
> Thank you for this comment. We are currently in the process of collecting a prospective, multi-center dataset (ClinicalTrials.gov [NCT04424030](https://clinicaltrials.gov/study/NCT04424030)) and look forward to validating our method in future work. In the meantime, we have noted this limitation in the discussion.
>
> > "...the dataset was obtained retrospectively from a single center; proposed network should undergo further validation in prospectively obtained, multicenter images."

---

### Official Review · Reviewer_9M83 · 2025-02-20

**Confidence:** 3
**Preliminary Rating:** 3
**Final Rating:** 4

**Summary:**

The article presents an end-to-end trained approach for segmenting the left ventricle from CCTA images while simultaneously learning the optimal rotation to align the CCTA volume with four standard cardiac planes: short axis (SAX), two chamber (2CH), three chamber (3CH), and four chamber (4CH). The proposed network follows a three-step process: (1) an initial coarse segmentation on a downsampled CCTA, (2) quaternion-based learning of the rotations from the segmentation network’s latent space, and (3) a fine segmentation on the SAX-oriented CCTA. The method is evaluated on a dataset of 313 CCTA scans, with an ablation study conducted to assess the contribution of each component.

**Strengths:**

- The article is clear and well written
- The proposed approach is validated on a large dataset of routinely acquired CCTA
- An ablation study is performed on each component of the approach
- The code to reproduce the results is available

**Weaknesses:**

- The paper lacks comparison with similar work, making it unclear if it improves on previous work
- Some method choices are not well justified.
- The ablation study lacks clarity, especially regarding cascading, end-to-end training, and Table 1’s "Hdn" column.

**Detailed Comments:**

The main weakness of this work is the lack of comparison with state-of-the-art approaches. While the authors mention the method by Chen et al. in their literature review, their critique of its shortcomings is neither well explained nor sufficiently justified:
- The authors argue that a limitation of the Chen et al. approach is its use of separate models to predict the standard cardiac planes rather than an end-to-end training framework. However, this is not inherently a drawback, and the reasoning behind this claim should be further elaborated.
- They also criticize the use of a single fully connected layer for predicting cardiac plane parameters, implying that this is a limitation. However, a smaller model can be advantageous if it achieves good performance with minimal computational cost. This aspect should be discussed more thoroughly.
- Additionally, the authors state that the Chen et al. approach does not reorient images into the cardiac coordinate system, suggesting that doing so could improve segmentation. However, this claim is made without supporting evidence, and its impact on segmentation accuracy should be demonstrated.
To support these claims, I recommend that the authors compare their method directly with the approach by Chen et al. Given the apparent similarities in architecture, implementing and evaluating the Chen et al. method should be feasible without excessive effort.

Some parts of the methods should be clarified. In particular, the dimensions of the volumes used by the authors are not clear :
- First the original CCTA are downsampled to 64x64x64 voxels 3mm isotropic volumes, then in the fine segmentation volumes, the original CCTA are resampled to 128×128×192 voxels 0.8mm isotropic volumes. This is not possible unless a croping is performed. This should be clarified.
- The inputs of the fine segmentation module are not clear. The description in the article is very succinct and Figure 1 is not clear (see next comment). The author should clarify this point

The authors state the their network was trained for 24 epochs. This is a very small yet precise number of epochs. The authors should explain why and how they set this number.

The ablation study is interesting, however it is unsufficiently described to be convincing. It is very not clear what is the impact on the proposed method of the absence of some component :
- The cascading component is never mentioned as part of the proposed method. This should be described depth before the ablation study.
- The end-to-end training is described, however, it is not clear what the authors did when removing it in the ablation study. Did they train separate networks ? If so which ones and how did they do ?
- The authors state that the width and number of hidden layers in the transformation module vary in the ablation study, seemingly referring to the "Hdn" column in Table 1. However, this column lacks a proper legend, and the numbers presented are not explained. A clearer description is needed to specify which numbers correspond to which architectural variations.

In the discussion, the authors state that their approach "combines concepts from spatial transformer networks and ...". However, it is unclear which aspects of their method incorporate principles from spatial transformer networks. This connection should be explicitly presented.

**Justification Of The Final Rating:**

Even though the lack of comparison with state-of-the-art approaches remains a major limitation, the authors have significantly improved their article, and the enhanced statistical analysis strengthens their results.
I therefore change my rating to weak accept.

**Justification Of The Preliminary Rating:**

The rating is primarily justified by the lack of comparison with existing methods, which significantly limits the ability to assess the true impact of the proposed approach.
In addition, some parts of the methodology and ablation study are not sufficiently detailed, making it difficult to fully understand the work conducted.

**Questions To Address In The Rebuttal:**

- The authors should clarify how the specific architectural choices of their method contribute to performance improvements over Chen et al., ideally by providing experimental results directly comparing the two approaches.

- The description of the proposed method should be more detailed and precise, addressing the ambiguities highlighted in the above detailed comments.

- The ablation study requires a clearer explanation, particularly regarding the cascading, end-to-end , and "Hdn" components.

---

> ### Author Response · Authors · 2025-03-08
> **Response to Reviewer 9M83**
>
> - R2-1: "...lack of comparison with state-of-the-art approaches."
>
> See R1-6, R0-1, R0-2, and R0-3 above.
>
> - R2-2: "...the authors mention the method by Chen et al. in their literature review, their critique of its shortcomings..."
>
> Please see R0-4 above.
>
> - R2-3: "The authors argue that a limitation of the Chen et al. approach is its use of separate models to predict the standard cardiac planes rather than an end-to-end training framework."
>
> We provided an ablation experiment demonstrating the benefit of end-to-end over sequential training of each module:
>
> > "Training the coarse segmentation, transformation, and fine segmentation modules separately degraded performance for all metrics."
>
> Regarding separate versus combined models, see R0-3 above.
>
> - R2-4: "They also criticize the use of a single fully connected layer..."
>
> Although not inherently a limitation (see R2-2 above), we demonstrate through an ablation experiment that removing the hidden layers from the fully connected branch comes with a performance penalty:
>
> > "Removing all hidden layers from the transformation module resulted in slightly lower centroid error (0.801 mm versus 0.805 mm) but degraded performance for all other metrics."
>
> - R2-4: "...the authors state that the Chen et al. approach does not reorient images into the cardiac coordinate system, suggesting that doing so could improve segmentation..."
>
> Please see R0-2, R1-4, and R1-6 above.
>
> - R2-5: "...the dimensions of the volumes used by the authors are not clear...This is not possible unless a croping is performed."
>
> That is correct and by design; one purpose of the coarse segmentation and transformation modules is to define a narrow field of view centered on the heart in the SAX coordinate system; the original image is resampled in this narrow field of view at high resolution before being passed into the second segmentation stage. We have clarified this in the methods:
>
> > "The proposed network architecture (Figure 1) consists of three end-to-end-trained modules: (a) a coarse segmentation module, which segments a large FOV, low resolution image, (b) a transformation module, which predicts the rotations corresponding to the standard cardiac planes, and (c) and a fine segmentation module, which segments a narrow FOV, high resolution image reformatted in the SAX coordinate system. The two segmentation modules are cascaded by resampling the coarse segmentation logits in the SAX coordinate system and providing these as additional channels to the input of the fine segmentation module."
>
> - R2-6: "The inputs of the fine segmentation module are not clear."
>
> We have reiterated the inputs to the fine segmentation module in the methods.
>
> - R2-7: "The authors state the their network was trained for 24 epochs. This is a very small yet precise number..."
>
> The approximate number of epochs was chosen by monitoring the training and validation losses for convergence. The specific number 24 was chosen because it is divisible by three, and convenient for the end-to-end training ablation experiment where we sequentially trained each of the three modules separately (given our desire to keep the total number of epochs fixed for all experiments in the interest of fair comparison).
>
> - R2-8: "The cascading component is never mentioned as part of the proposed method..."
>
> See R2-5 above.  Additionally, this has been reiterated in the results:
>
> > "Removing cascading (that is, providing only the resampled input image without the coarse segmentation logits to the fine segmentation module)..."
>
> - R2-9: "The end-to-end training is described, however, it is not clear what the authors did when removing it in the ablation study..."
>
> The results of sequentially training each module rather than training the network end-to-end are provided in Table 1. We have clarified the technique when discussing the ablation experiments:
>
> > "To test the effect of end-to-end training, we sequentially trained the coarse segmentation, transformation, and fine segmentation modules (8 epochs each, 24 epochs total), resulting in degraded performance for all metrics."
>
> - R2-10: "The authors state that the width and number of hidden layers in the transformation module vary in the ablation study, seemingly referring to the "Hdn" column in Table 1..."
>
> The table caption has been rewritten as below:
>
> > "...The Hn column indicates the number of features in each hidden layer of the transformation module (e.g., “64”: one 64-feature hidden layer; “64-64”: two 64-feature hidden layers; “–”: no hidden layers)..."
>
> - R2-11: "In the discussion, the authors state that their approach "combines concepts from spatial transformer networks and ..."..."
>
> In order to prioritize reporting additional experimental results given limited space, this claim has been removed.

---

> > ### Comment · Reviewer_9M83 · 2025-03-13
> >
> > I thank the authors for their responses. The ablation study has indeed been extended, and its presentation is now much clearer. They have addressed most of my comments; however, I still have one request:
> >
> > - Figure 1 should be improved by including the following details:
> >     - The dimensions of each input and output.
> >     - A description of \theta and the red line.
> >     - The names of each module (e.g., coarse segmentation, transformation module, etc.).
> >     - A depiction of supervision at each stage, for example, by showing each component of the loss.
> > These additions would significantly enhance the clarity of the method for the reader.
> >
> > Despite the improvements, I still have two major reservations regarding the article:
> >
> > 1. The experimental results remain somewhat inconclusive, as they do not strongly support the claim that the proposed approach is the best or that each component is essential. In particular, the performance of ablated versions—such as the model without cascading or without direct supervision—appears very close to that of the full method. Given the small differences in quantitative results, the significance of the drawn conclusions is questionable. To address this, the authors should report standard deviations and conduct statistical tests to determine which results are statistically significant, rather than simply highlighting values that are above or below their proposed approach.
> >
> > 2. The absence of a comparison with state-of-the-art methods remains a major limitation of the paper. While I acknowledge that the short rebuttal period may not have allowed the authors to implement the approach of Chen et al., this remains a critical issue.
> >
> > Overall, the paper is well written, and its clarity has improved, however, the presented results are not sufficiently convincing to fully support the proposed approach.

---

> > > ### Author Response · Authors · 2025-03-14
> > >
> > > Thank you once again for your valuable feedback.  Although the website does not allow me to upload a new revision during the discussion period, I have confirmed with the program organizers that I may provide further revisions in the camera-ready version in response to feedback during the discussion period.  As such:
> > >
> > > - Regarding Figure 1, we agree with all the feedback provided.  If fortunate enough for our paper to be accepted, we will provide all the requested revisions to the figure in the camera-ready version (unfortunately I do not seem to be able to embed images here).
> > > - We are also happy to provide standard deviations and significance testing in the camera-ready version.  Note that in adding the standard deviations, this table no longer fits on a single page, and so I have broken up the angle errors into one table (to be retained in the body) and the centroid error and dice scores into another (to be placed in the appendix). Please see below a simplified markdown version of the new tables. Also please note that, as I am unable to render color correctly on this submission website, in this simplified version I have indicated statistically significant improvements and degradations as up- and down-arrows, respectively.
> > > - With reference to these new tables (up- and down-arrows again indicating statistically significant differences), I think the following are the most important revised takeaways:
> > >   - Removing the transformation module hidden layer, removing end-to-end training, or removing indirect supervision all resulted in a statistically significant degradation in performance for all angle errors.
> > >   - Removing direct supervision did not result in a statistically significant change in the final angle error or any other metric; however, as discussed previously in the rebuttal (see R0-1 above): "removing direct supervision resulted in large errors in the intermediate quaternion rotations [...] Having accurate intermediate quaternions is useful should predicted planes require manual correction; therefore, we elect to provide direct supervision since doing so preserves the intermediate rotations and has minimal impact on the final planes."
> > >   - Removing the second stage segmentation module resulted in a statistically significant increase in two of the four angle errors.
> > >   - There was no statistically significant change in angle errors when training individual models on single planes compared to our model which predicts all planes simultaneously.  Given the difference in training time (14 hours for our proposed model versus 14x4=56hrs to train four separate models), we consider this to be a significant advantage of our method.
> > >   - We agree that the effect of attention gates, residual blocks, and cascading was overall equivocal.
> > >
> > > In the camera-ready version, we would be happy to refine our discussion to reflect these conclusions.

---

> > > > ### Author Response · Authors · 2025-03-14
> > > >
> > > > Table 1: Network performance, hyperparameter search, and ablation experiments. The effect of attention gates (An), residual blocks (Rs), cascading (Cd), indirect (Id) and direct (Dr) rotation supervision, end-to-end training (EE), the fine segmentation module (Fn), multiple vs single plane predictions (Pn), and the transformation module hidden layers (Hn) on
> > > > the angle errors for each plane are explored. The An, Rs, Cd, Id, Dr, EE, and Fn columns indicate whether the feature
> > > > was (“+”) or was not (“–”) employed. The Pn column indicates whether the model was trained to predict “All” planes
> > > > or a single (“SAX’’, “2CH’’, “3CH’’, or “4CH’’) plane. The Hn column indicates the number of features in each hidden
> > > > layer of the transformation module (e.g., “64”: one 64-feature hidden layer; “64-64”: two 64-feature hidden layers; “
> > > > –”: no hidden layers). Values are reported as “mean (standard deviation)”. Results significantly improved and worsened relative to the proposed network by two-tailed Student’s t-test (p<0.05) are indicated by up (&uarr;) and down (&darr;) arrows, respectively.
> > > >
> > > > | An | Rs | Cd | Id | Dr | EE | Fn | Pn | Hn | SAX                       | 2CH                       | 3CH                       | 4CH                       |
> > > > |----|----|----|----|----|----|----|----|----|---------------------------|---------------------------|---------------------------|---------------------------|
> > > > | +  | +  | +  | +  | +  | +  | +  | All  | 64      | 9.737 (5.754)                     | 9.858 (5.686)                     | 9.839 (5.159)                     | 9.565 (5.053)                     |
> > > > | +  | +  | +  | +  | +  | +  | +  | All  | 64-64   | 10.106 (5.603)                    | 10.040 (5.272)                    | &darr; 10.682 (5.406) | 9.401 (5.359)                     |
> > > > | +  | +  | +  | +  | +  | +  | +  | All  | 128     | 9.176 (5.813)                     | 9.096 (6.548)                     | &darr; 9.922 (5.502)  | &darr; 10.913 (7.467) |
> > > > | +  | +  | +  | +  | +  | +  | +  | All  | 128-128 | 9.126 (6.194)                     | 9.480 (5.412)                     | 9.030 (5.898)                     | 8.840 (5.920)                     |
> > > > | +  | +  | +  | +  | +  | +  | +  | All  | 256     | 9.708 (5.510)                     | 10.263 (4.963)                    | &darr; 10.683 (4.920) | &darr; 10.006 (5.498) |
> > > > | +  | +  | +  | +  | +  | +  | +  | All  | 256-256 | 9.961 (5.000)                     | 9.697 (5.231)                     | 9.667 (4.816)                     | 9.344 (5.240)                     |
> > > > | -- | +  | +  | +  | +  | +  | +  | All  | 128-128 | 9.886 (4.360)                     | 9.941 (4.890)                     | 9.718 (4.906)                     | 9.199 (4.770)                     |
> > > > | +  | -- | +  | +  | +  | +  | +  | All  | 128-128 | 9.316 (5.401)                     | 9.572 (5.473)                     | 9.786 (5.521)                     | 9.178 (6.029)                     |
> > > > | +  | +  | -- | +  | +  | +  | +  | All  | 128-128 | 10.149 (4.925)                    | 9.420 (6.004)                     | 10.034 (4.824)                    | 9.322 (4.921)                     |
> > > > | +  | +  | +  | -- | +  | +  | +  | All  | 128-128 | &darr; 10.775 (5.464) | &darr; 10.559 (6.212) | &darr; 10.564 (5.458) | &darr; 10.808 (5.395) |
> > > > | +  | +  | +  | +  | -- | +  | +  | All  | 128-128 | 9.363 (5.621)                     | 9.045 (5.847)                     | 9.131 (5.843)                     | 8.765 (5.901)                     |
> > > > | +  | +  | +  | +  | +  | -- | +  | All  | 128-128 | &darr; 12.959 (5.885) | &darr; 12.501 (7.025) | &darr; 12.837 (5.353) | &darr; 12.535 (5.697) |
> > > > | +  | +  | +  | +  | +  | +  | -- | All  | 128-128 | 11.068 (5.946)                    | 10.860 (6.561)                    | &darr; 10.352 (5.620) | &darr; 10.577 (6.322) |
> > > > | +  | +  | +  | +  | +  | +  | +  | SAX  | 128-128 | 8.707 (5.048)                     | --                                | --                                | --                                |
> > > > | +  | +  | +  | +  | +  | +  | +  | 2CH  | 128-128 | --                                | 8.734 (5.626)                     | --                                | --                                |
> > > > | +  | +  | +  | +  | +  | +  | +  | 3CH  | 128-128 | --                                | --                                | 8.994 (4.386)                     | --                                |
> > > > | +  | +  | +  | +  | +  | +  | +  | 4CH  | 128-128 | --                                | --                                | --                                | 8.965 (5.792)                     |
> > > > | +  | +  | +  | +  | +  | +  | +  | All  | --      | &darr; 11.559 (7.123) | &darr; 10.807 (6.659) | &darr; 10.401 (6.063) | 10.115 (5.983)                    |

---

> > > > > ### Author Response · Authors · 2025-03-14
> > > > >
> > > > > Table 2. Network performance, hyperparameter search, and ablation experiments. The effect of attention gates (An), residual blocks (Rs), cascading (Cd), indirect (Id) and direct (Dr) rotation supervision, end-to-end training (EE), the fine segmentation module (Fn), multiple vs single plane predictions (Pn), and the transformation module hidden layers (Hn)
> > > > > on the segmentation centroid (Ctd) and Dice scores for bloodpool (BP), myocardium (MC), and trabeculations (TB)
> > > > > are explored. The An, Rs, Cd, Id, Dr, EE, and Fn columns indicate whether the feature was (“+”) or was not (“–”)
> > > > > employed. The Pn column indicates whether the model was trained to predict “All” planes or a single (“SAX’’, “2CH’’,
> > > > > “3CH’’, or “4CH’’) plane. The Hn column indicates the number of features in each hidden layer of the transformation
> > > > > module (e.g., “64”: one 64-feature hidden layer; “64-64”: two 64-feature hidden layers; “–”: no hidden layers). Values are reported as “mean (standard deviation)”. Results significantly improved and worsened relative to the proposed network by two-tailed Student’s t-test (p<0.05) are indicated by up (&uarr;) and down (&darr;) arrows, respectively.
> > > > >
> > > > > | An | Rs | Cd | Id | Dr | EE | Fn | Pn | Hn | SAX                       | 2CH                       | 3CH                       | 4CH                       |
> > > > > |----|----|----|----|----|----|----|----|----|---------------------------|---------------------------|---------------------------|---------------------------|
> > > > > | +  | +  | +  | +  | +  | +  | +  | All  | &darr; 2.246 (1.116) | 0.955 (0.007)                      | 0.930 (0.016)                      | &uarr; 0.814 (0.030) |
> > > > > | +  | +  | +  | +  | +  | +  | +  | All  | &darr; 2.184 (1.077) | 0.956 (0.007)                      | 0.929 (0.014)                      | 0.810 (0.032)                      |
> > > > > | +  | +  | +  | +  | +  | +  | +  | All  | 0.853 (0.554)                    | 0.955 (0.007)                      | &darr; 0.926 (0.016)   | 0.808 (0.029)                      |
> > > > > | +  | +  | +  | +  | +  | +  | +  | All  | 0.805 (0.521)                    | 0.955 (0.008)                      | 0.928 (0.016)                      | 0.808 (0.029)                      |
> > > > > | +  | +  | +  | +  | +  | +  | +  | All  | &darr; 1.941 (1.043) | 0.955 (0.007)                      | &uarr; 0.930 (0.016) | 0.809 (0.032)                      |
> > > > > | +  | +  | +  | +  | +  | +  | +  | All  | 0.820 (0.605)                    | &darr; 0.954 (0.007)   | &darr; 0.921 (0.016)   | &darr; 0.801 (0.032)   |
> > > > > | -- | +  | +  | +  | +  | +  | +  | All  | &darr; 1.917 (1.000) | 0.954 (0.007)                      | 0.928 (0.016)                      | &uarr; 0.813 (0.029) |
> > > > > | +  | -- | +  | +  | +  | +  | +  | All  | 0.811 (0.575)                    | &darr; 0.954 (0.007)   | 0.927 (0.014)                      | 0.805 (0.031)                      |
> > > > > | +  | +  | -- | +  | +  | +  | +  | All  | 0.787 (0.562)                    | 0.957 (0.008)                      | 0.931 (0.017)                      | &uarr; 0.817 (0.029) |
> > > > > | +  | +  | +  | -- | +  | +  | +  | All  | 0.781 (0.629)                    | 0.954 (0.007)                      | &darr; 0.925 (0.016)   | 0.805 (0.031)                      |
> > > > > | +  | +  | +  | +  | -- | +  | +  | All  | 0.772 (0.492)                    | 0.955 (0.007)                      | 0.929 (0.015)                      | 0.812 (0.029)                      |
> > > > > | +  | +  | +  | +  | +  | -- | +  | All  | &darr; 1.302 (0.739) | &darr; 0.927 (0.013)   | &darr; 0.905 (0.017)   | &darr; 0.717 (0.041)   |
> > > > > | +  | +  | +  | +  | +  | +  | -- | All  | 0.800 (0.599)                    | &uarr; 0.958 (0.008) | 0.926 (0.019)                      | &uarr; 0.834 (0.029) |
> > > > > | +  | +  | +  | +  | +  | +  | +  | SAX  | 0.795 (0.565)                    | &darr; 0.950 (0.009)   | 0.927 (0.017)                      | &darr; 0.799 (0.040)   |
> > > > > | +  | +  | +  | +  | +  | +  | +  | 2CH  | 0.864 (0.562)                    | &darr; 0.951 (0.008)   | 0.927 (0.016)                      | &darr; 0.801 (0.038)   |
> > > > > | +  | +  | +  | +  | +  | +  | +  | 3CH  | 0.830 (0.629)                    | 0.954 (0.008)                      | 0.929 (0.016)                      | 0.810 (0.034)                      |
> > > > > | +  | +  | +  | +  | +  | +  | +  | 4CH  | 0.764 (0.539)                    | &darr; 0.950 (0.010)   | 0.926 (0.015)                      | &darr; 0.797 (0.042)   |
> > > > > | +  | +  | +  | +  | +  | +  | +  | All  | 0.801 (0.445)                    | &darr; 0.954 (0.007)   | 0.926 (0.014)                      | 0.805 (0.033)                      |

---

> > > > > > ### Author Response · Authors · 2025-03-15
> > > > > >
> > > > > > Followup:  I have added an updated version of Figure 1 (the schematic of the network architecture) to the README on the [landing page of the GitHub repository](https://github.com/sudomakeinstall/2025-midl-ccta-plane-prediction/tree/main?tab=readme-ov-file), since openreview.net does not allow embedding images directly into comments.  The updated version of the figure (a) specifies the matrix size and resolution at every stage, (b) labels the coarse and fine segmentation modules, and (c) labels the inputs to each term of the loss function.  If we are fortunate enough for this paper to be accepted, I will continue to refine this figure in the camera-ready version; however, I did want to demonstrate some progress toward addressing the reviewer's concerns before the end of the discussion period.

---

### Official Review · Reviewer_rzvk · 2025-02-20

**Confidence:** 4
**Preliminary Rating:** 4
**Final Rating:** 4

**Summary:**

This paper proposes a two-stage segmentation approach for left ventricle (LV) segmentation in cardiac CTA scans. The first stage performs a rough segmentation of the LV on low-resolution scans while also predicting cardiac planes by estimating quaternions for the short-axis, two-chamber, three-chamber, and four-chamber views. The image and segmentation are then transformed into the short-axis view, and the second stage refines the segmentation using a high-resolution network. The approach is clinically relevant, as both segmentation and cardiac plane prediction are essential for cardiovascular analysis. Additionally, this method could be extended to segment other cardiac structures, such as the atria and right ventricle. While the paper includes an ablation study on the network architecture, it lacks a comparison to related work, which would strengthen the evaluation.

**Strengths:**

- The paper is well-written and easy to follow.
- Cardiac plane prediction is clinically relevant and contributes to standardizing imaging views.
- The combination of coarse-to-fine segmentation with intermediate plane prediction and resampling is an interesting idea, even though the overall two-step segmentation strategy is not new.
- The limitations section is thorough and acknowledges potential areas for improvement.

**Weaknesses:**

Minor weaknesses:

- As noted in the limitations, an analysis of inter- and intra-rater variability for quaternion angles would be valuable.
- There is no explanation for why the short-axis view was chosen for the second segmentation stage. Understanding whether this choice was made based on empirical results or clinical considerations would be helpful.
- The paper does not include an ablation study on the loss function L_q​, which consists of both indirect and direct supervision terms. Further justification for this choice would be beneficial.

Major weaknesses:

- It is unclear whether the cardiac plane prediction and resampling step significantly improve segmentation performance. A comparison to using only the high-resolution segmentation network with the original images would clarify its contribution. It would be helpful to confirm whether this is already evaluated in the ablation study (i.e., the model "without cascading"). Some additional clarification on this point would strengthen the analysis.
- The dataset section lacks details about how the segmentation labels and plane angles were obtained.
- The paper does not include a comparison with related work, such as the method by Chen et al. (Chen 2021b), which is mentioned in the introduction. Given the similarities between the approaches, a direct comparison would be beneficial.

**Detailed Comments:**

- Why was the short-axis view selected for the second segmentation step? Was this choice based on empirical evidence, or is it primarily driven by clinical relevance? Did the authors experiment with other views for this step?
- The loss function Lq​ comprises both an indirect and a direct supervision term. Could the authors elaborate on why this combination was chosen? Would direct supervision alone not suffice? An ablation study on the impact of each term in the loss function would provide additional insight.

**Justification Of The Final Rating:**

The authors addressed all my concerns in their rebuttal. I vote to accept this paper, as it is of interest to the community and the method seems sound. I agree with reviewer 9M83 that a comparison with state of the art would be ideal, however I also understand the author's problems with reproducing the results by Chen et al.

**Justification Of The Preliminary Rating:**

- plane prediction and resampling is an important topic in cardiac image analysis
- method seems to work quite well and is simple
- some clarifications are needed and additional experiments would improve the paper.

**Questions To Address In The Rebuttal:**

Could the authors clarify what "removing cascading" entails in the ablation study? Does this correspond to using only the high-resolution segmentation network without the initial low-resolution step and plane prediction?
Please provide additional details on how the segmentation labels and quaternion angles were obtained.

---

> ### Author Response · Authors · 2025-03-08
> **Response to Reviewer rzvk**
>
> - R1-1: "an analysis of inter- and intra-rater variability for quaternion angles would be valuable."
>
> We agree and plan to report inter- and intra-rater variability in future work, but are unable to provide this analysis within the time constraints of the rebuttal period. This is noted as a limitation:
>
> > "it would be useful to quantify intra- and inter-observer variability in standard cardiac plane angles in order to contextualize the angle errors observed in our network."
>
> - R1-2: "There is no explanation for why the short-axis view was chosen for the second segmentation stage. Understanding whether this choice was made based on empirical results or clinical considerations would be helpful."
>
> Thank you--as you suggest, this decision was clinically motivated. We have expanded upon our reasoning in the methods section:
>
> > "The SAX plane is chosen for the second segmentation stage because, unlike the long axis planes, the SAX plane is routinely reviewed as a stack from base to apex and maps directly onto the bullseye plots commonly used to display downstream analyses such as wall thickness, wall thickening, and segmental strain."
>
> We additionally provide new experiments in which separate models are trained to predict each individual cardiac plane. In these models, the images are reformatted into whichever plane was predicted, since the SAX plane is not necessarily available, with no compelling performance improvement in any of these models compared to our proposed model.
>
> - R1-3: "The paper does not include an ablation study on the loss function L_q​, which consists of both indirect and direct supervision terms. Further justification for this choice would be beneficial."
>
> Thank you for this suggestion--the experiments have been added as suggested.  Please see R0-1 above.
>
> - R1-4: "It is unclear whether the cardiac plane prediction and resampling step significantly improve segmentation performance. A comparison to using only the high-resolution segmentation network with the original images would clarify its contribution. It would be helpful to confirm whether this is already evaluated in the ablation study (i.e., the model "without cascading"). Some additional clarification on this point would strengthen the analysis."
>
> Thank you for this suggestion--this experiment has been added as suggested.  Please see R0-2 above. Additionally, regarding the second half of the reviewer's comment: "cascading" refers to whether the features learned by the first stage segmentation network are passed as an additional input to the second segmentation stage alongside the resampled input image (see Figure 1 from Wu, et al, for the version used in our architecture); the input image is resampled in the predicted short axis coordinate system; if the first stage segmentation logits are also provided, the network is said to be cascaded. These architectural features are independent and are evaluated by separate ablation experiments. We have clarified this point throughout the paper (see R2-5, R2-6, and R2-8 below).
>
> - R1-5: "The dataset section lacks details about how the segmentation labels and plane angles were obtained."
>
> We have added additional details regarding segmentation labels and plane angles to the methods section:
>
> > "Initial myocardial and bloodpool segmentations were obtained using a previously described network (Kong et al., 2021) and trabeculations were separated from the bloodpool by thresholding. These initial segmentations were manually corrected (AM, AH, and KW) using ITK-Snap (version 3.8.0). Standard cardiac planes were defined by a cardiologist with fellowship training in cardiac imaging and 10 years of experience (TG)."
>
> - R1-6: "The paper does not include a comparison with related work, such as the method by Chen et al. (Chen 2021b), which is mentioned in the introduction. Given the similarities between the approaches, a direct comparison would be beneficial."
>
> We agree that a direct comparison would be beneficial.  Unfortunately, we were unable to directly run the code referenced in that manuscript due to library incompatibility issues resulting from incomplete specification of the library versions. We additionally attempted to re-implement Chen, et al., but in training the re-implementation found the network to be unstable, resulting in the introduction of `NaN`s with no network output. We were unable to fully debug this issue within the rebuttal period, and so have noted the lack of direct comparison as a limitation in the discussion:
>
> > "Last, although we explore through ablation experiments many of the features which distinguish our network from the most closely related work (Chen et al., 2021b), a direct head-to-head comparison would be valuable."

---

> > ### Comment · Reviewer_rzvk · 2025-03-11
> >
> > Thanks for a thorough rebuttal and replying to all of my comments.
> > I think the authors did a great job providing more ablation experiments and also mentioning limitations in their discussion.
> >
> > Do you have an explanation why the centroid error improves when removing indirect or direct supervision?

---

> > > ### Author Response · Authors · 2025-03-11
> > >
> > > Thanks to you as well!  We are extremely appreciative of your thoughtful criticisms, which we feel have substantively improved the paper.
> > >
> > > Thanks also for the followup question, which I would address in two ways:
> > >
> > > - Note that the weights of the encoder path of the first stage segmentation network contribute to both the centroid (calculated from the coarse segmentation) and the quaternion rotations. By removing direct or indirect quaternion supervision, this component of the task is simplified, theoretically allowing the network to focus a greater portion of its attention on the segmentation component of the task.
> > > - However, I would be cautious not to over-interpret this change.  First, the magnitude was very small at ~0.2mm (for context, the first stage segmentation network has an input pixel size of 3x3x3mm). Second, the change was not statistically significant for either the direct (p=0.288) or indirect (p=0.609) supervision ablation experiments by two-tailed Student’s t-test (though we did not think it appropriate to report statistical significance in the paper since the analysis was informal, without prospective power analysis, multiple comparisons correction, etc).
> > >
> > > I hope this addresses your question but am of course happy to discuss more!

---

### Author Rebuttal · Authors · 2025-03-08

**Rebuttal:**

Thank you to all reviewers for your time in reviewing our manuscript, and for your thoughtful and constructive feedback. We present a substantive revision incorporating your suggestions to the greatest degree possible within the time constraints of the 7-day rebuttal period. In particular, we have added three sets of ablation experiments in response to reviewer comments:

- Alternatively removing indirect and direct supervision.
- Exploring the utility of the second segmentation stage.
- Exploring the utility of predicting all standard cardiac planes versus separate networks to predict each cardiac plane.

Additionally, our critique of prior work is re-framed in the introduction and the method has been clarified in several areas.  Further details will be provided using the "Official Comment" feature shortly.

Again, sincerest thanks to the reviewers for their time and feedback.  If there are any points of interest or areas requiring further clarification, I would be very happy to engage during the upcoming discussion period.

**Supporting Material:**

/attachment/99abecd0c6af71efab1409130d6b8f7e968b1e87.pdf

---

> ### Author Response · Authors · 2025-03-08
> **Rebuttal: General Responses to All Reviewers**
>
> Thank you to all reviewers for your time in reviewing our manuscript, and for your thoughtful and constructive feedback. We highlight the most consequential revisions here, with additional detailed responses to individual reviewers below.
>
> - R0-1: We have added ablation experiments alternatively removing indirect and direct supervision.
>
> The following has been added to the results:
>
> > "Removing indirect supervision of the quaternion rotations degraded performance in terms of the angle errors for all standard cardiac planes and for all Dice scores; centroid error was slightly lower (0.781 mm vs 0.805 mm). Removing direct supervision of the quaternion rotations degraded performance in terms of the angle errors for the SAX and 3CH planes and the bloodpool Dice; performance was slightly improved in terms of centroid error (0.772 mm vs 0.805 mm), angle error for the 2CH ($9.045^\circ$ vs $9.480^\circ$) and 4CH ($8.765^\circ$ vs $8.840^\circ$) planes, myocardial Dice (0.929 vs 0.928), and trabeculation Dice (0.812 vs 0.808). However, removing direct supervision resulted in large errors in the intermediate quaternion rotations: $182.659^\circ$, $182.066^\circ$, $183.317^\circ$, $183.185^\circ$, and $183.596^\circ$ vs $6.611^\circ$, $6.837^\circ$, $7.077^\circ$, $6.876^\circ$, and $6.304^\circ$ for $Q_{BLN}$, $Q_{\Delta SAX}$, $Q_{\Delta 2CH}$, $Q_{\Delta 3CH}$, and $Q_{\Delta 4CH}$, respectively. Having accurate intermediate quaternions is useful should predicted planes require manual correction; therefore, we elect to provide direct supervision since doing so preserves the intermediate rotations and has minimal impact on the final planes."
>
> - R0-2: We have added an ablation experiment exploring the utility of the second segmentation stage.
>
> The following has been added to the results:
>
> > "To test the utility of the second segmentation stage, we removed the second segmentation stage, instead inputting the full field of view, high-resolution images to the first segmentation stage (requiring a reduction in the number of features in the first stage U-Net by a factor of 4 due to GPU memory constraints). Doing so degrades performance in terms of all angle errors and myocardial Dice; performance was slightly improved in terms of centroid error (0.800 mm vs 0.805 mm), bloodpool Dice (0.958 vs 0.955), and trabeculation Dice (0.834 vs 0.808)."
>
> The following has been added to the discussion:
>
> > "Fourth, we report that removing the second stage segmentation module degrades cardiac plane prediction without substantially changing segmentation performance; the dominant effect on plane error is somewhat counterintuitive and deserves further investigation."
>
> - R0-3: We have added an ablation experiment exploring the utility of predicting all standard cardiac planes versus separate networks to predict each cardiac plane.
>
> The following has been added to the results:
>
> > "To test the utility of predicting all standard cardiac planes in a single network, we trained four separate networks, each predicting a single cardiac plane. The SAX, 2CH, and 3CH angle errors slightly improved ($8.707^\circ$ vs $9.126^\circ$ SAX; $8.734^\circ$ vs $9.480^\circ$ 2CH; $8.994^\circ$ vs $9.030^\circ$ 3CH) and the 4CH angle error slightly worsened ($8.965^\circ$ vs $8.840^\circ$); differences in centroid error and Dice scores were also small."
>
> Given that the differences in plane errors were small ($<1^\circ$ difference in all cases), we do not feel that the additional complexity, time, and expense of training separate networks for each plane is justified.
>
> - R0-4: We have re-framed our discussion in the literature review as follows:
>
> > "The most closely related work (Chen et al., 2021b) describes a method to predict 2CH, 3CH, and 4CH planes from a U-Net bottleneck; however, several architectural and training decisions deserve further exploration. (a) Separate models are trained to predict each cardiac plane, multiplying training time, but without comparing to a single unified model. (b) Their network is trained using a multi-stage approach, but without comparing to end-to-end training. (c) Regarding the fully connected network branched from the bottleneck, no experiments are reported exploring the effect of hidden layers (either their presence, number, or width) on performance. (d) Promising modifications to the U-Net such as attention gates and residual blocks are not explored. (e) Having learned the transformation parameters, it is reasonable to question whether performance could be improved by segmenting the reformatted images in a second stage; however, this was not investigated."
>
> - R0-5: In order to accommodate responses to reviewer comments, certain training and implementation details have been moved to an appendix.

---

> > ### Author Response · Authors · 2025-03-12
> > **Reproducibility**
> >
> > FYI: The [GitHub](https://github.com/sudomakeinstall/2025-midl-ccta-plane-prediction/) repository has now been updated to include all the additional experiments which were requested in the reviews.  Changes from the initial submission are given [here](https://github.com/sudomakeinstall/2025-midl-ccta-plane-prediction/commit/ee7453c6ae2b3f5e96eb893edaede61dbb3b1776).

---

### Meta-Review · Area_Chair_GKzW · 2025-03-21

**Recommendation:** Accept (Poster)
**Confidence:** 5

**Metareview:**

This paper proposes a two-stage segmentation approach for left ventricle (LV) segmentation in cardiac CTA scans. The article is clear and well written. The proposed approach is validated on a large dataset of routinely acquired CCTA. An ablation study is performed on each component of the approach.  The code to reproduce the results is available. The enhanced statistical analysis strengthens their results